# Students’ Mental Health, Well-Being, and Loneliness during the COVID-19 Pandemic: A Cross-National Study

**DOI:** 10.3390/healthcare10060996

**Published:** 2022-05-27

**Authors:** Tore Bonsaksen, Vivian Chiu, Janni Leung, Mariyana Schoultz, Hilde Thygesen, Daicia Price, Mary Ruffolo, Amy Østertun Geirdal

**Affiliations:** 1Department of Health and Nursing Sciences, Faculty of Social and Health Studies, Inland Norway University of Applied Sciences, 2418 Elverum, Norway; 2Department of Health, Faculty of Health Studies, VID Specialized University, 4603 Sandnes, Norway; 3Faculty of Health and Behavioural Science, The University of Queensland, St. Lucia, QLD 4072, Australia; vivian.chiu@uq.net.au (V.C.); j.leung1@uq.edu.au (J.L.); 4Faculty of Health and Life Sciences, Northumbria University, Newcastle upon Tyne NE1 8ST, UK; mariyana.schoultz@northumbria.ac.uk; 5Department of Occupational Therapy, Prosthetics and Orthotics, Faculty of Health Sciences, Oslo Metropolitan University, 0130 Oslo, Norway; hilde.thygesen@oslomet.no; 6Department of Health, Faculty of Health Studies, VID Specialized University, 0370 Oslo, Norway; 7School of Social Work, University of Michigan, Ann Arbor, MI 48109, USA; daiciars@umich.edu (D.P.); mruffolo@umich.edu (M.R.); 8Department of Social Work, Faculty of Social Sciences, Oslo Metropolitan University, 0130 Oslo, Norway; amyoge@oslomet.no

**Keywords:** COVID-19, cross-national study, mental health, pandemic, social distancing

## Abstract

During the COVID-19 pandemic, much research has been devoted to assessing mental health in a variety of populations. Students in higher education appear to be particularly vulnerable to experiencing reduced mental health. The purpose of the study was to assess whether higher education students experienced poorer mental health compared to the general population and examine the factors associated with students’ mental health during the COVID-19 pandemic. A cross-national sample of students (*n* = 354) and non-students (*n* = 3120) participated in a survey in October/November 2020. Mental health outcomes among students and non-students were compared with independent *t*-tests. Multiple linear regression analysis and general linear estimation were used to assess the impact of student status on mental health outcomes while adjusting for sociodemographic factors. Students reported poorer mental health than non-students. The difference in mental health between students and non-students was bigger for participants aged 30 years or older. More social media use was associated with poorer mental health outcomes. In conclusion, students had poorer mental health than the wider population. Aspects of life as a student, beyond what can be attributed to life stage, appears to increase mental health problems.

## 1. Introduction

There is substantial evidence that the outbreak of the coronavirus disease (COVID-19) and the social disruptions it has caused are associated with psychological symptoms such as anxiety, depression, post-traumatic stress disorder and stress among adults worldwide [1,2,3,4]. While findings from general population studies may not generalize to the adult student population, research indicates that students have been prone to experience stress and subsequent mental health problems during the COVID-19 pandemic [5,6,7].

Epidemiological studies conducted before and during the COVID-19 pandemic have consistently found a high prevalence of anxiety and mood disorders among university or college students [8,9]. University years are also a peak period for the onset of these disorders [9]. Stressors driven by academic demands, shift in social roles and instability of educational and occupational opportunities leave university students vulnerable to mental health problems, such as anxiety and depression [8,9,10]. Early evidence suggests that the COVID-19 pandemic has exacerbated preexisting mental health problems among students due to academic disruption, unstable learning environment and reduced peer connection [11]. While changes in study or work patterns and general increased uncertainty may have affected most groups during the pandemic, a systematic review found that student status was a significant risk factor for experiencing depressive symptoms during COVID-19 compared with employed and retired people [2]. The effects of age and student status may converge, as most students are relatively young. Therefore, students may experience more emotional distress due to school closures, lack of social events and more childcaring responsibility but possibly also due to lower study efficiency with remote online courses [12]. Students have also been known to have low help-seeking behavior [13,14], and social stigma and health-related fear during the pandemic may further discourage students from engaging with mental health services. Their access to mental health services may also have been limited due to the pandemic restrictions.

The COVID-19 pandemic has another feature in the digital era, where people rely heavily on digital platforms such as social media for communication and information. Social media has a dual nature during the pandemic. Live broadcasting of news and the interactive commenting function on social media allow users to engage in dialogues and pass on information faster than ever. It also fosters social connection when face-to-face communication is restricted. However, previous studies have established the association between excessive disaster media exposure and poor mental health outcomes [15,16]. At the same time, the circulating information can be fused with rumors and misinformation, which can be difficult to differentiate [17]. The constant stream of coverage about the COVID-19 outbreak, along with the spread of misinformation, creates fear, anxiety and stress [17,18,19]. Moreover, social media use has been shown to relate differently to loneliness among people in different age groups, with more social media use being linked with more loneliness among those of younger age [20]. However, there is a need to disentangle the influence of the student role from that of younger age and to examine whether predictors of students’ mental health, well-being and loneliness are similar to those in the general population.

The aim of this study was to examine (i) whether students experienced poorer mental health, poorer well-being and more loneliness compared to the rest of the population and (ii) the factors associated with mental health, well-being, and loneliness among students during the COVID-19 pandemic.

## 2. Methods

### 2.1. Design and Procedures

A cross-sectional survey was conducted in Norway, USA, UK and Australia. Social media platforms (i.e., Facebook and Twitter) were used to distribute the online survey between 24 October and 29 November 2020. During the time the survey was available, Norway experienced between 2000 and 4000 new cases weekly, while the UK experienced between 105,000 and 175,000 and the USA experienced between 560,000 and 1,280,000 new cases weekly. Australia experienced less than 100 new cases weekly during the whole data collection period [21]. A landing site for the survey was established at the researchers’ universities; OsloMet—Oslo Metropolitan University, Norway; University of Michigan, USA; Northumbria University, UK; and the University of Queensland, Australia. Each country had their own project lead, according to the country-specific rules. The survey was developed by the researchers in two languages simultaneously: Norwegian and English, and was based on a previous survey conducted by the research group in the early phase (April 2020) of the pandemic outbreak [18,19,22,23,24] Language and cultural differences were considered during the survey development process. This means that the Norwegian phrasing of each item would convey the same content as the corresponding English item, while considering the different grammatical structures and nuances in the culturally embedded meaning of words allowed us to use the phrase that would most effectively convey the meaning in each of the languages.

### 2.2. Participants

To be included in the study, participants were required to be 18 years of age or older; to understand the language in which the survey was presented (Norwegian or English) and to be a citizen of one of the relevant countries at the time of the survey (Norway, USA, UK or Australia).

### 2.3. Measures

#### 2.3.1. Sociodemographic Characteristics

Sociodemographic variables included age group (18–24 years, 25–29 years and 30 years or older); gender (male versus female) and cohabitation (living with spouse or partner versus not). Participants who indicated that they were not employed, or only partially employed, were asked to further specify their status, of which one was ‘I am a student’. In this study, those responding ‘I am a student’ were classified as student, while the rest of the sample were classified as not being a student.

#### 2.3.2. COVID-19 Variables

Relating to COVID-19 infection, the participants were asked whether they had been infected by COVID-19. The question was answered with yes or no. Participants were also asked whether they were, or had been, in self-isolation or quarantine during the pandemic. A combined variable was constructed, distinguishing between those who had experienced self-isolation and/or quarantine during the pandemic and those who had not experienced any of these measures.

#### 2.3.3. Social Media Use

The participants were asked to indicate the amount of time they had spent on social media on a typical day during the last month. In line with the work of Ellison and co-workers [25], the response options were less than 10 min, 10–30 min, 31–60 min, 1–2 h, 2–3 h and more than three hours. Based on the distribution of scores on this variable, social media use was subsequently dichotomized into low-level use (1–2 h daily or less; 57% of the sample) versus high-level use (2–3 h daily or more; 43% of the sample).

#### 2.3.4. Mental Health

General Health Questionnaire 12 (GHQ-12) is a self-reported measure of mental health [26,27]. Its validity across samples and contexts has been demonstrated in a large number of studies in general adult, clinical, work and student populations [27,28,29,30,31]. Six items of the GHQ-12 are phrased positively (e.g., ‘able to enjoy day-to-day activities’), while six items are phrased as a negative experience (e.g., ‘felt constantly under strain’). For each item, the person indicates the degree to which he or she has experienced the item content during the past two weeks (‘less than usual’, ‘as usual’, ‘more than usual’ or ‘much more than usual’). Items are scored between 0 and 3, and positively formulated items are recoded prior to analysis. As a result, the GHQ-12 scale score range is 0–36, with higher scores indicating poorer mental health (more psychological distress). In the whole sample, Cronbach’s *α* for the GHQ-12 was 0.91, identical to the measure found in the student sample.

#### 2.3.5. Well-Being

The PsychoSocial Well-being (PSW) scale measures an individual’s psychological experience of well-being with ten items. The scale includes five positive and five negative statements, and all item scores range between 1 (= highest well-being) and 5 (= lowest well-being). Cronbach’s alpha was 0.91 in the whole sample and 0.89 in the restricted student sample. Validity tests of the PSW scale provided evidence to support the use of a one-factor solution [24], as used in this study, and this is consistent with previous validation studies [32].

#### 2.3.6. Loneliness

Loneliness was measured with the Loneliness Scale [33]. The scale consists of six statements, all of which rated from 0 (totally disagree) to 4 (totally agree). It was designed to measure two different aspects of loneliness: ‘emotional loneliness’ and ‘social loneliness’. Previous factor-analytic studies have found two latent factors underlying the six items, and the items should therefore be used as two separate scales, reflecting the two different aspects of loneliness [33,34]. In the whole sample, internal consistencies (Cronbach’s α) were 0.70 and 0.88 for the emotional loneliness and social loneliness scales, respectively. In the restricted student sample, Cronbach’s α was 0.60 for emotional loneliness and 0.86 for social loneliness.

### 2.4. Statistical Analysis

SPSS software was used to analyze the data [35]. Differences in proportions between students and the rest of the population by sociodemographic characteristics were analyzed with the chi-square test. Group differences on continuous measures (mental health, well-being and social and emotional loneliness) were analyzed with independent *t*-tests. Adjusted associations between student status and each of the outcomes were examined with linear regression models. In these regression analyses, the independent variables were age group, gender, living with spouse or partner and student status. To examine whether any associations between student status and the outcomes were moderated by age, additional analyses entered the interaction term student status × age group. General linear model analysis was used to estimate the mean outcome levels for categories of the moderator variable while adjusting for the sociodemographic factors.

Then, we restricted the sample to only include the students and examined the predictors of mental health, well-being and social and emotional loneliness among the students. These regression models included age group, gender, living with spouse or partner, having been in quarantine or in self-isolation during the pandemic, having been infected with COVID-19 and the level of social media use. Additional analyses examined potential differences between students from different countries. Statistical significance was set at *p* < 0.05, and effect sizes were reported as Cohen’s *d* (independent *t*-tests), standardized *β* weights (linear regression) [36] and partial *η*^2^ [37]. Guided by Newman’s [38] advice to use all available data, missing values were managed with case-wise deletion, resulting in *n* varying between analyses.

### 2.5. Ethics

The data collected in this study were anonymous. The researchers adhered to all relevant regulations in their respective countries concerning ethics and data protection. The study was approved by OsloMet (20/03676) and the regional committees for medical and health research ethics (REK; ref. 132066) in Norway, reviewed by the University of Michigan Institutional Review Board for Health Sciences and Behavioral Sciences (IRB HSBS) and designated as exempt (HUM00180296) in the USA, by Northumbria University Health Research Ethics (HSR1920-080) in UK and HSR1920-080 2020000956 in Australia.

## 3. Results

### 3.1. Participants

The sociodemographic characteristics of the students (*n* = 354) and the rest of the population (*n* = 3120) are displayed in Table 1. The largest proportion of students came from the USA (*n* = 178, 50.3%), followed by the UK (*n* = 95, 26.8%), Norway (*n* = 61, 17.3%) and Australia (*n* = 20, 5.6%). Compared to the general population, students had higher proportions in the younger age groups, a higher proportion of women and a higher proportion not living with a spouse or partner.

### 3.2. Mental Health Differences between Students and the General Population

Table 2 displays the results from the independent *t*-tests of differences in mental health, well-being and social and emotional loneliness between the students and the general population. Compared with the general population, the students had significantly poorer mental health and well-being and had greater emotional loneliness. The largest difference was shown for psychosocial well-being. The difference in social loneliness was not statistically significant.

Table 3 displays the results from the regression analyses, where differences between students and the general population are adjusted for age, gender and living with spouse or partner. The analyses showed that a student status was significantly associated with poorer mental health even when adjusting for the sociodemographic covariates, while associations between student status and well-being and loneliness were nonsignificant. The additional interaction analysis showed that student status × age was significantly associated with mental health (*p* < 0.01), indicating that the association between being a student and experiencing poorer mental health varied by age. In the analyses of predictors of well-being and loneliness, the interaction term was not statistically significant.

The analysis of predictors of mental health in the samples split by age (18–29 years versus 30 years or older) is shown in Table 4. Student status was borderline associated with mental health among those aged 18–29 years, with poorer mental health estimates for students (*M*: 19.6, 95% CI: 18.7–20.4) than for non-students (*M*: 18.4, 95% CI: 17.7–19.1, *p* = 0.05). Among those aged 30 years or older, students (*M*: 17.6, 95% CI: 16.1–19.1) had significantly poorer mental health estimates compared to non-students (*M*: 15.7, 95% CI: 15.4–16.0, *p* = 0.02). Significant effects of gender and living with a partner were absent among those aged 18–29 years, while among those aged 30 years or higher, mental health was poorer for women (*p* < 0.001) and for those living without a spouse or partner (*p* = 0.001) compared to their male and cohabitating counterparts. All effect sizes were small (partial *η*^2^ ranging 0.00–0.03).

### 3.3. Factors Associated with Mental Health, Well-Being, and Loneliness among Students

Table 5 displays the results from the regression analyses restricted to the student participants only. Higher social media use was found to be associated with poorer mental health, poorer well-being and higher levels of emotional loneliness, with small-to-moderate effect sizes. In addition, living with a spouse or partner was related with higher well-being and lower emotional loneliness, and having been in isolation or quarantine was related to higher emotional loneliness.

### 3.4. Differences between Countries

Considering the cross-national sample used in this study, we analyzed cross-country differences with regards to the students’ mental health, well-being and loneliness. Of the four countries, Australia (*n* = 20) did not have a student sample large enough to be meaningfully compared with students in the other countries. Thus, we included dummy variables to distinguish between students from Norway, the USA and the UK in subsequent regression analyses. Adjusting for all variables (age, gender, living with spouse or partner, self- isolation or quarantine, COVID-19 infection and social media use), UK students had significantly poorer mental health (*β* = −0.15, *p* = 0.03), poorer well-being (*β* = −0.17, *p* = 0.01) and higher levels of emotional loneliness (*β* = −0.16, *p* = 0.02) compared to students from the USA. Compared to students from Norway, the USA students had greater social loneliness (*β* = 0.17, *p* = 0.03), while UK students had poorer well-being (*β* = 0.18, *p* = 0.047) and greater social loneliness (*β* = 0.26, *p* = 0.01) and emotional loneliness (*β* = 0.25, *p* = 0.01).

## 4. Discussion

This study found that, during the COVID-19 pandemic, students experienced poorer mental health, lower well-being and greater loneliness, compared with the general population. After adjusting for the sociodemographic characteristics, mental health among students remained significantly worse compared to that of the general population. The results suggest that there are aspects of life as a student—other than the general sociodemographic composition of the student group—that links with poorer mental health. The mental health gap between students and non-students was bigger for older than for younger persons. The study also demonstrated that students who spent more time using social media experienced poorer mental health outcomes compared to students spending less time on social media. Finally, students’ mental health was found to differ between countries and was notably worse among students in the UK compared to students in Norway and in the USA.

Mental health problems are prevalent among students, with one-third of students having experienced one or more mental health disorders at some point across the countries [9]. It has also been shown that the COVID-19 pandemic has negatively influenced students’ perceived stress, substance use and mental health [5,6,7,8]. Much in line with these reports, our findings showed that students experienced more mental health symptoms than the general population, even when controlling for age, gender and cohabitation. Thus, the COVID-19 pandemic may have widened the mental health gap between students and the rest of the population. However, one should note that we did not conduct a study about specific mental disorders among students, and we did not use measures to tap into specific symptomatology. Instead, we compared students and non-students with regards to their self-reported mental health, psychosocial well-being and loneliness. Notably, the gap between students and non-students was bigger for those of higher age, possibly reflecting higher demands related to the complex interplay between study, work, childcare and other family obligations being placed on older students during the pandemic [39].

While most students were young, the age difference did not fully account for the detected mental health differences between students and the rest of the population. Reasons for the additional vulnerabilities among students may include the academic disruption instigated by the pandemic and the measures implemented against it. Such disruptions have caused concerns about school closures, academic delays and disruption to learning routines. These are common reasons for students to feel worried and anxious [11]. Even though perceptions of loneliness were found to be fairly similar between students and non-students, another reason for students’ poorer mental health may concern perceptions of social isolation during the pandemic situation. While many academic institutions have completely moved their course materials online to minimize academic disruption, physical supportive networks of supervisors and peers have been difficult. The social connection to other people may be particularly critical for young people transitioning from high school to university, as it takes extra effort to adjust to the new environment and be accepted as part of the community. Students who fail to fulfil their social needs are susceptible to experience negative emotions and substantially reduce their satisfaction with their university learning experiences [40], which potentially compromises their academic performance.

In the general population sample, higher age, male gender and living with a partner were generally associated with favorable psychological outcomes. Contrastingly, in the student sample, living with a spouse or partner was the only sociodemographic variable that was associated with better outcomes. Living with a partner has generally been associated with better mental health outcomes [41,42,43], and students appear to be no exception. However, the COVID-19 literature has also reported poorer mental health [43,44,45] and more concerns [46] among younger persons and among women, while this was not found among the students in our study. Possibly, the expected mental health differences related to age and gender are not as established among young students in higher education as they are later in life. Alternatively, a lack of variation on the age and gender variables (most students were women and relatively young) may explain why age and gender were not significantly associated with the mental health outcomes among the students.

Among the students, more time spent using social media was associated with poorer mental health and well-being and more emotional loneliness. While these findings are in line with those of several studies conducted during and before the COVID-19 pandemic [24,47,48,49,50,51], we note that the strength of the associations detected within the student sample appears to be more substantive than the comparable associations found in the general populations studied e.g., [20,52]. Social media is by far more widely adopted among adolescents and young adults, compared to those in the older age groups [53]. Their use may therefore also exert a stronger influence on young people, as social media is often an integral part of young people’s social presence and lifestyle. Thus, social media use may be more strongly linked to mental health among young students than in the general population. Possibly, students and young people in general may have a stronger inclination towards social comparison and to seek affirmation from others, and social media may be used for these purposes [54]. In the case of negative social comparison (e.g., “others are more successful than me”) or unfulfilled needs for affirmation (e.g., “nobody liked the picture I posted”), social media use may fuel emotional distress instead of relieving it. However, depending on the direction of a possible causal link, social media use may impact negatively on students’ mental health, or students with more mental health problems may be inclined to spend more time on social media. Notably, the latter view may also be argued from several theoretical standpoints [55]. For example, people with higher levels of depressive symptoms may be more inclined to use social media frequently to alleviate negative feelings [56] or to seek social affirmation [57]. They may also be more inclined to develop harmful behavioral patterns more generally due to poorer self-regulatory skills [58]. Thus, interpreting associations between social media use and mental health is complex and should be done bearing in mind the diversity of the explanatory models that exist.

Mental health, well-being and loneliness were found to be worse among students in the UK than among students in Norway and the USA. Before the outbreak of COVID-19, the UK and USA were both found to be well-prepared for a pandemic [59]. Nonetheless, both countries have had large numbers of infections and deaths during the pandemic crisis, while the corresponding numbers for Norway have been relatively low [21]. However, it appears that, some time into the pandemic, the UK shifted towards implementing more restrictive policies to tackle the outbreak while the USA, despite allowing states to make individual plans, maintained their more open policies at the federal level. The more restrictive policies (i.e., self-isolation and ‘stay at home’ orders) in the UK during the later stage of the pandemic may contribute to explaining the poorer mental health among the students in the UK. While Norway has also had strict national restrictions implemented during the pandemic crisis, Norway appears to have been more effective in reducing the level of virus transmission. Restrictions in Norway may therefore have been shorter and milder than the policies implemented in the UK. Thus, the nature and duration of pandemic restrictions and, also, their perceived effects on disease transmission in the population seem to be highly relevant for an understanding of the pandemic’s mental health consequences.

### 4.1. Implications for Student Mental Health Support

To support students’ mental health during the coronavirus pandemic, several courses of action may be considered. Establishing a virtual student community is easy and economically viable. The goal of such a community is to restore the sense of belonging when physical social activities are restricted. Key elements required to facilitate engagement in virtual student communities are a supportive environment, well-functioning communication systems and regular content updates [60].

Sufficient resources to meet students’ needs for mental health services should be ensured [61]. Counsellors may be telehealth-trained to offer off-site intervention in addition to the existing face-to-face services to accommodate ongoing social distancing practices. Web-based interventions may also encourage students to access the service, as it provides a higher confidentiality. Within the study programs, the education on mental health across universities may be strengthened to enhance student awareness of mental health symptoms and knowing when and where to seek help.

Several studies during the COVID-19 pandemic have pointed to financial concerns as one of several possible reasons for mental health problems in the general population [43,44,62]. Fear of losing one’s job, or having problems with finding a job, may be particularly widespread among young people who are more often in a vulnerable economic situation. Enhancing financial support so students have a more stable living situation may therefore indirectly contribute to support students’ mental health. Older students with children and family obligations may be helped if offered childcare support and other means to sustain a balance between the conflicting demands of education, work and family life.

### 4.2. Study Limitations

The cross-sectional design of the study may have only captured a transient observation of mental health symptoms rather than long-lasting consequences from the COVID-19. Data were collected under the COVID-19 context, but we did not have previous data pre-COVID-19 for comparison. Moreover, the data were collected in several different countries and over an extended period of time, and social restrictions, including periods of lockdown, have varied over time, as well as between and within countries. Thus, the participants’ experiences of the pandemic situation would vary substantially, and terms such as ‘self-isolation’ might embed different meanings to different people. For example, we do not know the duration of the self-isolation period for those who reported that they had been self-isolating.

The study design also prohibits us from making causal links between the variables under study, although some mechanisms are suggested. It is possible that the detected associations between student status and poorer mental health outcomes are confounded by other variables not accounted for in the study. For example, we have no information about the participants’ income level. Income is likely different between students and non-students, and a higher income has been found to be strongly associated with better mental health [63]. The possibility to compare students from the different countries was limited, due to small cell sizes when stratified by country. Much caution should be exercised when interpreting these results. The mental health of students probably differs between countries and universities, but the factors associated with poorer mental health are likely to be consistent.

The question probing about employment status, which led to the follow-up question to which participants could indicate ‘student’ status, may have been ambiguous. For example, PhD students in the USA are not salaried and would likely identify as ‘students’, whereas they are salaried and might be just as likely to identify as ‘employed’ in Norway. Possibly, this ambiguity might explain the large differences between the sizes of the student samples from the four countries. While the study provided evidence of poorer mental health among students compared to non-students in the general population, we have been left to speculate about the reasons for this difference. Although most university-aged students use social media, our sample may not be representative of the general population. More research is needed to address this issue.

## 5. Conclusions

In this study, we found that students had poorer mental health compared with non-students in the general population. The poorer mental health among students is not fully explained by their younger age and living arrangement composition. Students’ mental health may be particularly challenged by academic, as well as social, disruptions experienced during the pandemic. More social media use may be both cause and consequence of poorer mental health among students, and future research may further address the causal relationship between social media use and mental health. Students’ mental health differed between countries, possibly indicating that restrictive policies to counter the pandemic at the national level has had substantial negative mental health effects among students affected by the policies.

## Figures and Tables

**Table 1 healthcare-10-00996-t001:** Sociodemographic characteristics of students and in the general population.

Characteristics	Students		General Population		
	*n*	%	*n*	%	*p*
Age group					<0.001
18–24 years	176	62.4	106	37.6	
25–29 years	65	18.5	286	81.5	
30 years or older	113	4.0	2728	96.0	
Gender					<0.05
Male	64	8.3	707	91.7	
Female	279	11.0	2267	89.0	
Living with spouse/partner					<0.001
Yes	97	4.8	1943	95.2	
No	257	17.9	1177	82.1	

Note. Statistical tests are chi-square tests. Proportions are within categories. On the gender variable, 157 cases (4.5%) were removed due to missing or nonbinary responses.

**Table 2 healthcare-10-00996-t002:** The students’ and the general populations’ mental health, well-being and loneliness.

Variables	Students	General Population	Difference	Effect Size	
	*M* (*SD*)	*M* (*SD*)	*M* (*SE*)	Cohen’s *d*	*p*
GHQ 12	19.3 (7.2)	16.1 (6.7)	3.2 (0.4)	0.46	<0.001
PSW	3.2 (0.8)	2.7 (0.9)	0.4 (0.1)	0.59	<0.001
Emotional loneliness	7.4 (2.7)	5.9 (2.9)	1.5 (0.2)	0.53	<0.001
Social loneliness	4.6 (2.9)	4.4 (3.1)	0.1 (0.2)	0.07	0.46

Note. Statistical tests are the independent *t*-test. GHQ 12 is the mental health measure, while PSW is the well-being measure. Higher values indicate poorer mental health, poorer well-being and more social and emotional loneliness. Missing values were found among 12.7% of the participants, and these were excluded from the analyses.

**Table 3 healthcare-10-00996-t003:** Associations with mental health outcomes.

Independent Variables	Mental Health	Well-Being	Social Loneliness	Emotional Loneliness
Age group	−0.13 ***	−0.13 ***	0.07 **	−0.18 ***
Gender	0.13 ***	0.10 ***	−0.05 **	0.10 ***
Living with spouse/partner	−0.07 ***	−0.18 ***	−0.16 ***	−0.12 ***
Student status	0.05 *	0.03	0.02	0.02
**Explained variance**	**5.5% *****	**8.0% *****	**2.7% *****	**7.6% *****

Note. Table contents are standardized *β* weights. On the independent variables, higher values indicate higher age, female gender, living with spouse partner and being student. Of the dependent variables, higher values indicate poorer mental health, poorer well-being and more social and emotional loneliness. Missing values were found in between 10.5% and 11.7% of the participants, and these were excluded from the analyses. * *p* < 0.05, ** *p* < 0.01 and *** *p* < 0.001.

**Table 4 healthcare-10-00996-t004:** Predictors of mental health (GHQ scores) by age group.

Independent Variables	*F*	*p*	Partial *η*^2^
18–29 years *(n* = 611)			
Gender	0.86	0.35	0.00
Living with spouse/partner	2.50	0.11	0.00
Student status	3.87	0.05	0.01
30+ years *(n* = 2421)			
Gender	69.84	<0.001	0.03
Living with spouse/partner	11.27	0.001	0.01
Student status	5.86	0.02	0.00

Note. Missing values were found in 12.7% of the participants; these were excluded from the analyses.

**Table 5 healthcare-10-00996-t005:** Associations with mental health outcomes among students (*n* = 286–288).

Independent Variables	Mental Health	Well-Being	Social Loneliness	Emotional Loneliness
Age group	−0.03	−0.00	0.08	−0.07
Gender	−0.02	−0.02	0.03	−0.01
Living with spouse/partner	−0.06	−0.15 *	−0.10	−0.14 *
Self-isolation/quarantine	0.11	0.08	0.11	0.14 *
Infected by COVID-19	−0.05	−0.10	−0.01	−0.07
Social media use	0.17 **	0.23 ***	0.05	0.17 **
**Explained variance**	**5.5% ***	**9.4% *****	**2.6%**	**9.5% *****

Note. Table contents are standardized *β* weights. On the independent variables, higher values indicate higher age, female gender, living with spouse partner, having been in self-isolation/quarantine, having been infected by COVID-19 and higher levels of social media use. On the dependent variables, higher values indicate poorer mental health, poorer well-being and more social and emotional loneliness. Missing values were found in 18.6% of the student participants, and these were excluded from the analyses. * *p* < 0.05, ** *p* < 0.01 and *** *p* < 0.001.

## Data Availability

There is no associated dataset.

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
