# Peer review of "Students’ Mental Health, Well-Being, and Loneliness during the COVID-19 Pandemic: A Cross-National Study"

_healthcare, 2022, doi:10.3390/healthcare10060996_

Round 1
Reviewer 1 Report
This is an interesting study examining factors associated with students' mental health during the COVID-19 pandemic. The paper is well-written with good sample size. I appreciate the fact that the authors are very in interpreting their results. This is commendable. I believe that the paper will be able to contribute to the literature. I have several comments to help the authors to improve the manuscripts further:
- The introduction is clear and well-written. However, I think it will be important for the authors to provide more justifications and rationale on why they expect students to experience poorer mental outcomes compared to the rest of the population. I understand that some justifications have been provided in the second paragraph. But one can argue that academic/work disruption, increased uncertainty and reduced peer connection may also applicable with the non-students too. So, more elaboration will be helpful to strengthen the theoretical justification of the current paper.
- The paper mentioned that listwise deletion was used for missing data. It will be useful to report the percentage of data that was removed. A brief justification on the decision to use listwise deletion will be helpful too. Relevant paper:
Newman, D. A. (2014). Missing data: Five practical guidelines. Organizational Research Methods, 17(4), 372-411. - It will be useful to elaborate how language and cultural differences were considered during the survey development process. If back translation technique is used, it will be useful to mention in the method section. Relevant paper:
Chen, H. Y., & Boore, J. R. (2010). Translation and back‐translation in qualitative nursing research: methodological review. Journal of Clinical Nursing, 19(1‐2), 234-239. - It will be useful for the authors to report zero-order correlations of all main variables in the study.
- Regarding the finding on negative correlations between social media use and mental health outcomes, it will be important for the authors to discuss the possibility of reverse causation in the discussion. While social media may affect mental health, it is equally possible that social media is the antecedent rather than the cause of lowered well-being and mental health outcomes. The authors did mention briefly about this issue but I think it will be nice to elaborate further to provide a more balanced view on this important issue. Relevant paper: Quek, F. Y., Tng, G. Y., & Yong, J. C. (2021). Does social media use increase depressive symptoms? A reverse causation perspective. Frontiers in Psychiatry, 12, 335.
- A bit of justifications on the choice of covariates included in the main model will be useful. There are other demographic factors such as household income or other SES indicators that may differ between student and non-student sample and are strongly associated with well-being outcomes too.
- In addition, in the method section, it will be helpful for the authors to briefly mention the severity of the pandemic outbreak in October and November 2020 in Norway, USA, UK, and Australia to provide some contextual information for future readers.
Author Response
Reviewer 1 (R1): This is an interesting study examining factors associated with students' mental health during the COVID-19 pandemic. The paper is well-written with good sample size. I appreciate the fact that the authors are very in interpreting their results. This is commendable. I believe that the paper will be able to contribute to the literature. I have several comments to help the authors to improve the manuscripts further.
Authors: Thank you for the feedback.
R1: The introduction is clear and well-written. However, I think it will be important for the authors to provide more justifications and rationale on why they expect students to experience poorer mental outcomes compared to the rest of the population. I understand that some justifications have been provided in the second paragraph. But one can argue that academic/work disruption, increased uncertainty and reduced peer connection may also applicable with the non-students too. So, more elaboration will be helpful to strengthen the theoretical justification of the current paper.
Authors: We have elaborated on this issue in the revised introduction section, with references added.
R1: The paper mentioned that listwise deletion was used for missing data. It will be useful to report the percentage of data that was removed. A brief justification on the decision to use listwise deletion will be helpful too. Relevant paper: Newman, D. A. (2014). Missing data: Five practical guidelines. Organizational Research Methods, 17(4), 372-411.
Authors: We have mentioned that missing values were handled with casewise (not listwise) deletion, resulting in n varying between analyses (see 2.5 analysis section). In the revised manuscript, we have included a rationale for the employed method and included the suggested reference. The magnitude of missing values has been reported beneath each of the tables in the revised manuscript.
R1: It will be useful to elaborate how language and cultural differences were considered during the survey development process. If back translation technique is used, it will be useful to mention in the method section. Relevant paper: Chen, H. Y., & Boore, J. R. (2010). Translation and back‐translation in qualitative nursing research: methodological review. Journal of Clinical Nursing, 19(1‐2), 234-239.
Authors: The survey was developed simultaneously in two languages (English and Norwegian). More information has been added about the survey development process (see revised section 2.1).
R1: It will be useful for the authors to report zero-order correlations of all main variables in the study.
Authors: While this information can be easily produced, we believe it may be a sidetrack to the presentation of results pertaining to the study aim: to examine i) whether students experienced poorer mental health, poorer well-being, and more loneliness compared to the rest of the population; and ii) factors associated with mental health, well-being, and loneliness among students during the COVID-19 pandemic. We believe that the results we presented are more informative in addressing the aims, compared to zero-order correlations, because we present standardized beta weights that take into consideration all variables of interest. However, if the editor believes that this information is important to include, we could do so.
R1: Regarding the finding on negative correlations between social media use and mental health outcomes, it will be important for the authors to discuss the possibility of reverse causation in the discussion. While social media may affect mental health, it is equally possible that social media is the antecedent rather than the cause of lowered well-being and mental health outcomes. The authors did mention briefly about this issue but I think it will be nice to elaborate further to provide a more balanced view on this important issue. Relevant paper: Quek, F. Y., Tng, G. Y., & Yong, J. C. (2021). Does social media use increase depressive symptoms? A reverse causation perspective. Frontiers in Psychiatry, 12, 335.
Authors: We have expanded on this issue in the discussion section and included several references, including the one suggested.
R1: A bit of justifications on the choice of covariates included in the main model will be useful. There are other demographic factors such as household income or other SES indicators that may differ between student and non-student sample and are strongly associated with well-being outcomes too.
Authors: We believe the inclusion of age, gender, cohabitation, employment and living area are well-established indicators that relate to mental health outcomes. It is possible that the detected associations between student status and poorer mental health outcomes are influenced by other variables not accounted for in the study, e.g. income, as suggested by the reviewer. We have noted this issue in the study limitations section (section 4.2), along with a relevant reference.
R1: In addition, in the method section, it will be helpful for the authors to briefly mention the severity of the pandemic outbreak in October and November 2020 in Norway, USA, UK, and Australia to provide some contextual information for future readers.
Authors: This information has been added in the Methods section (section 2.1), as suggested.
Reviewer 2 Report
I reviewed the article titled: “Students’ mental health, well-being, and loneliness during the COVID-19 pandemic: a cross-national survey” and I found it very interesting and well-prepared. Bonsaksen et al. analysed the mental health of students in four countries and identified the factors that might have been linked with poorer mental health during the COVID-19 pandemic. The statistical analysis and the explanation of the results seems reasonable and reliable, although I found some points that should be verified. In my opinion the article might be published in Healthcare after minor revision. Please, check and answer the following points:
Introduction:
[40] addictive substance use
Materials and Methods:
[85] The authors indicated that cultural differences were considered during the survey. What cultural differences were taken into account? Could you elaborate it?
[103] Did the authors take into account lockdown restrictions that had been applied that time in the surveyed countries? How many participants answered “yes” for the question about self-isolation and whether the lockdown restrictions and the length of the lockdown had influence on the answer for this question?
[145] What tool was used for statistical analysis?
Discussion:
[328] Could you indicate how many days the lockdown last in each country?
References:
[407] The number of self-citations of the first author is 13 out of 53 (24,5%). In my opinion the number of self-citations is above the limit. As far as I can see all the cited works are recent, so I leave the decision to the editor.
Author Response
Reviewer 2 (R2): I reviewed the article titled: “Students’ mental health, well-being, and loneliness during the COVID-19 pandemic: a cross-national survey” and I found it very interesting and well-prepared. Bonsaksen et al. analysed the mental health of students in four countries and identified the factors that might have been linked with poorer mental health during the COVID-19 pandemic. The statistical analysis and the explanation of the results seems reasonable and reliable, although I found some points that should be verified. In my opinion the article might be published in Healthcare after minor revision.
Authors: Thank you for the feedback.
R2: Please, check and answer the following points: Introduction: [40] addictive substance use
Authors: We have removed the term ‘substance use’ from the manuscript as it has low relevance to our study.
R2: Materials and Methods: [85] The authors indicated that cultural differences were considered during the survey. What cultural differences were taken into account? Could you elaborate it?
Authors: We have elaborated on the point, as also indicated by Reviewer 1. See revised section 2.1.
R2: [103] Did the authors take into account lockdown restrictions that had been applied that time in the surveyed countries? How many participants answered “yes” for the question about self-isolation and whether the lockdown restrictions and the length of the lockdown had influence on the answer for this question?
Authors: We did not take lockdown restrictions into account, for several reasons: The survey was open for more than one month, meaning that restrictions in a given area might vary during the data collection period. Moreover, some restrictions were national, whereas others were local, depending on the level of transmission in the given area. We realize that the participants’ mental health would likely vary according to their personal experience with COVID-19, including lockdown periods. This has been noted in the study limitations section. 1921 participants (55%) had experienced self-isolation and/or quarantine. We have no information about the length of lockdown or the specifics of social restrictions during the data collection period (see revised study limitations section).
R2: [145] What tool was used for statistical analysis?
Authors: SPSS was used to analyze the data. This has been stated in the revised analysis description, section 2.4.
R2: Discussion: [328] Could you indicate how many days the lockdown last in each country?
Authors: Unfortunately, we cannot, there were local differences and we do not have this data. This has been noted in the study limitations section.
R2: References: [407] The number of self-citations of the first author is 13 out of 53 (24,5%). In my opinion the number of self-citations is above the limit. As far as I can see all the cited works are recent, so I leave the decision to the editor.
Authors: We believe all citations, including self-citations, are relevant. However, if the editor believes that some references should be deleted, we will naturally do so.
Reviewer 3 Report
Thank you for the opportunity to review an interesting topic „Students’ mental health, well-being, and loneliness during the COVID-19 pandemic: a cross-national survey“.
Major concerns
L 40: It is necessary to specify which mood disorders are being written about. It is necessary to justify why mood and anxiety disorders cover a wide range of conditions that fall under the umbrella of mental health disorders. Additionally, when it comes to disorders such as anxiety, it is necessary to explain or fix specific disorders or symptomatology in a sample of students.
L 42-44: there is a lack of information about the fact that academic strain as a stressor leads to psychosocial distress. A psychological stress as a trigger can lead to the development of anxiety, and the risk of depression.
Table 1: it is necessary (in cross tabulations: 2 x 2) to recalculate data using Fisher's exact test.
Table 2: data on means and their differences must be complemented. Simply adding a p value is not enough.
The authors associate social media use with poorer student mental health. It would be optimal to explain this relationship in the Discussion Unit by using mechanism of psychology.
The Conclusions could be rewritten by generating data and providing constructive findings. It is irrational to rewrite goals in Conclusions, etc.
There is also necessary to reject the comparisons by country due to the small number of students in subgroups. In addition, I would recommend that authors review the manuscript carefully and assess whether logical wheel errors have been made. Also provide that the design requirements of the cross-sectional study have been complied with.
Minor concerns
A survey can be qualitative, quantitative or mix methods. I would therefore recommend using the term “study” in the title of the manuscript.
The abstract lacks a backgruond sentence.
L 168-174: I would recommend writing this information in the back matter „Institutional Review Board Statement“.
L 94: I would recommend that the Authors do not write “gender” throughout the entire text of the manuscript, but write “sex”. The „gender“ term is also used more broadly to denote a range of identities that do not correspond to established ideas of male and female.
Author Response
Reviewer 3 (R3): Thank you for the opportunity to review an interesting topic „Students’ mental health, well-being, and loneliness during the COVID-19 pandemic: a cross-national survey“.
Authors: Thank you for the feedback.
R3: Major concerns. L 40: It is necessary to specify which mood disorders are being written about. It is necessary to justify why mood and anxiety disorders cover a wide range of conditions that fall under the umbrella of mental health disorders. Additionally, when it comes to disorders such as anxiety, it is necessary to explain or fix specific disorders or symptomatology in a sample of students.
Authors: We have not conducted a study about specific mental disorders, and we have not used measures to tap into specific symptomatology. Instead, we have compared students and non-students with regards to their self-reported mental health, psychosocial well-being, and loneliness. Please see the revised second paragraph of the Discussion section.
R3: L 42-44: there is a lack of information about the fact that academic strain as a stressor leads to psychosocial distress. A psychological stress as a trigger can lead to the development of anxiety, and the risk of depression.
Authors: We have included anxiety and depression as examples of mental health problems that may arise from psychological stress, and we have added more references; see revised introduction section.
R3: Table 1: it is necessary (in cross tabulations: 2 x 2) to recalculate data using Fisher's exact test.
Authors: As shown in Table 1, no cells have less than 64 participants. Given that Fisher’s Exact test is used in cases where the expected cell count is very small (i.e., less than five participants), we would argue that our use of the standard Chi Square test is appropriate.
R3: Table 2: data on means and their differences must be complemented. Simply adding a p value is not enough.
Authors: As shown in Table 2, the table content included mean scores and the corresponding standard deviations. We complemented them with estimates of mean differences between groups with the corresponding standard errors and effect sizes related to the differences between groups, in addition to p-values.
R3: The authors associate social media use with poorer student mental health. It would be optimal to explain this relationship in the Discussion Unit by using mechanism of psychology.
Authors: We appreciate this comment and have included a more detailed account of possible psychological mechanisms that may explain a relationship between social media use and poorer mental health. Please see the revised discussion section.
R3: The Conclusions could be rewritten by generating data and providing constructive findings. It is irrational to rewrite goals in Conclusions, etc.
Authors: According to the reviewer’s suggestion, we have removed the study aims in the beginning of the Conclusion section. In the revised manuscript, the conclusion section starts by stating the generated findings.
R3: There is also necessary to reject the comparisons by country due to the small number of students in subgroups. In addition, I would recommend that authors review the manuscript carefully and assess whether logical wheel errors have been made. Also provide that the design requirements of the cross-sectional study have been complied with.
Authors: As we removed the small subsample of students from Australia prior to analysis, the group sizes varied between 61 and 178 (see section 3.1). We found that relatively small effect sizes (i.e., 0.15) were statistically significant (see section 3.4), indicating sufficient statistical power for the analysis. Obviously, the student sample is much smaller than the non-student sample and interpreting the results pertaining to the cross-country differences between students should be done with much caution. Please see the revised study limitations section. We are not sure what is meant by “assess whether logical wheel errors have been made”. The study had a cross-sectional design, with limitations as outlined in the study limitations section.
R3: Minor concerns: A survey can be qualitative, quantitative or mix methods. I would therefore recommend using the term “study” in the title of the manuscript.
Authors: We have incorporated this suggestion in the manuscript title.
R3: The abstract lacks a backgruond sentence.
Authors: We appreciate this comment, and we have added two sentences to clarify the background in the abstract section.
R3: L 168-174: I would recommend writing this information in the back matter „Institutional Review Board Statement“.
Authors: We included that the researchers adhered to all relevant regulations in their respective countries concerning ethics and data protection and that we have ethics board approval. We will clarify any other required standard procedures in the production process.
R3: L 94: I would recommend that the Authors do not write “gender” throughout the entire text of the manuscript, but write “sex”. The „gender“ term is also used more broadly to denote a range of identities that do not correspond to established ideas of male and female.
Authors: We would like to keep ‘gender’ as is, in line with current social research and how our data was collected.
Round 2
Reviewer 1 Report
The authors have addressed all my concerns and comments well. I appreciate their efforts.
Reviewer 3 Report
Thank you for the opportunity to re-review an interesting topic „Students’ mental health, well-being, and loneliness during the COVID-19 pandemic: a cross-national study“. Authors greatly improved the text, however this improvement was not enough. A lot of issues remained unclear. Only after a clear description of the aims of this study and of the statistical approach will I be able to accept the paper.
Major conerns
I would recommend using the word „study“ throughout the manuscript.
Another important weakness is that the Authors provided data on the fact that they measured mental health (L 128: „General Health Questionnaire 12 (GHQ-12) is a self-report measure of mental health“). GHQ-12 scores show only problems related that psychological distress. Psychological distress is not an indicator of mental health. There are the 4 types of mental health: anxiety disorders, personality disorders, psychotic disorders (such as schizophrenia) eating disorders. Therefore, it seems necessary to clarify the entire manuscript, including the conclusions.
Furthermore, it is not clear why the authors carry out linear regression analysis with the categorical (nominal) variables such as gender, living with spouse/partner, student status. In this case, logistical regression analysis must be done. Table 3 shows standardized β weights, but neither standard errors nor 95% CI are found in the document. It is also realistic to describe regression models by providing R2 values.
In addition, in the text of the manuscript, the Authors wrote “L 231: the analysis of predictors of mental health...”. It should be noted that the authors carried out only a single cross-sectional study in which concepts such as "predictors" cannot exist.
The weakest part of the manuscript is the presentation of the Methods. As can be seen from the study design authors conducted a cross-sectional study. Thus, I suggest the authors to use STROBE checklist in reporting their cross-sectional study. Please, add the STROBE checklist as a supplementary material and cite it through the main text. You can download the checklist from this link https://www.strobe-statement.org/. Also, in accordance with the STROBE checklist, I will kindly request the authors to update the Unit of Methodology in the paper.